# Enhancing secondary school students' science process skills through guided inquiry-based laboratory activities in biology

Ashebir Mekonnen Chengere [1]*, Beyene Dobo Bono[2], Samuel Assefa Zinabu[3], Kedir Woliy Jilo[4]

1 Department of Biology, College of Natural and Computational Science, Hawassa University, Hawassa, Ethiopia, 2 Department of Biology, College of Natural and Computational Science, Hawassa University, Hawassa, Ethiopia, 3 School of Education, College of Education, Hawassa University, Hawassa, Ethiopia, 4 Department of Biology, College of Natural and Computational Science, Hawassa University, Hawassa, Ethiopia

* ashum2009@gmail.com

## Abstract

Science process skills (SPS) are vital for enhancing student engagement, critical thinking, and academic achievement in science education. However, Ethiopian secondary schools often rely on traditional, rote-based laboratory methods that hinder SPS development. This study examined the effect of Guided Inquiry-Based Laboratory Experiments enriched Instructional (GIBLEI) approach on improving students' SPS in biology. GIBLEI promotes active, inquiry-based learning, encouraging students to investigate, hypothesize, experiment, and draw conclusions. By shifting from passive observation to hands-on exploration, GIBLEI addresses limitations of traditional methods, fostering deeper understanding, problem-solving skills, and reducing educational disparities in science classrooms. In this quasi-experimental study, two biology classes from selected schools were randomly assigned to experimental (EG, N=46) and control groups (CG, N=29). The EG received GIBLEI-based instruction for eight weeks, focusing on inquiry-based laboratory activities that require students to investigate, hypothesize, and draw conclusions. The CG, meanwhile, received traditional lab instruction with a focus on observation and confirmation of set procedures. Data on SPS development were gathered using essay tests scored with rubrics. Welch's t-test revealed that post-test SPS scores in the EG were significantly higher than those in the CG, with a large effect size (82%), demonstrating GIBLEI's effectiveness. ANCOVA further confirmed that the improvement was attributable to the GIBLEI approach rather than initial group differences. The Wilcoxon signed-rank test showed significant SPS improvement within the EG from pre-test to post-test, underscoring the approach's effectiveness over time. Additionally, an independent samples t-test indicated no significant gender differences in SPS within the EG, suggesting that GIBLEI benefits both male and female students equally. The findings highlight GIBLEI as a promising tool to foster SPS, supporting its integration into biology curricula to enhance student engagement, skill acquisition, and equal learning outcomes across genders.

**Data availability statement:** All relevant data are within the manuscript and its Supporting Information files.

**Funding:** The author(s) received no specific funding for this work.

**Competing interests:** The authors have declared that no competing interests exist.

# 1. Introduction

Science Process Skills (SPS) are critical competencies that enable students to think critically, solve problems, and carry out scientific investigations effectively. These skills are essential for understanding science and technology, as well as for addressing various personal and societal challenges [1]. In science education, developing SPS is fundamental for equipping students with the tools necessary to navigate complex scientific problems. These skills can be broadly categorized into two types: Basic Science Process Skills (BSPS) and Integrated Science Process Skills (ISPS). Both categories are essential at all educational levels to build a solid foundation for scientific knowledge and practical problem-solving [2]. BSPS are introductory skills that serve as the foundation for developing more advanced skills. They help students observe their surroundings, communicate ideas effectively, measure accurately, classify objects or data, infer conclusions from observations, and predict outcomes based on evidence. These skills are necessary for constructing new knowledge and preparing students to engage with more complex scientific concepts. In contrast, ISPS involve more advanced skills that are typically used in scientific problem-solving and experimental investigations. These skills require higher-level cognitive abilities and a deeper understanding of scientific methods. Common ISPS indicators include identifying variables, formulating hypotheses, designing and conducting experiments, interpreting data, drawing conclusions, and building models [3].

The Ethiopian Ministry of Education recognizes the need to strengthen STEM (Science, Technology, Engineering, and Mathematics) education to support national development [4]. However, secondary school students, particularly in biology, struggle to develop essential SPS due to teacher-centered instructional methods and traditional laboratory practices. While laboratory experiments are vital for conceptual understanding and skill development in biology, chemistry, and physics, the rigid, step-by-step structure of traditional laboratory instruction limits opportunities for independent inquiry [5]. In these traditional settings, teachers dictate objectives and methods, leaving students to passively follow instructions rather than engage in active problem-solving [6]. This approach prioritizes factual recall over higher-order skills such as hypothesis formulation, experimental design, data interpretation, and drawing conclusion. Beyond instructional challenges, inadequate resources, including laboratory equipment, teaching materials, and well-equipped classrooms, hinder practical experimentation, forcing teachers to rely on theoretical explanations [7]. This scarcity restricts experiential learning, essential for developing SPS such as data analysis and hypothesis testing [8]. Additionally, large class sizes reduce opportunities for individualized instruction, limiting students' engagement in hands-on activities and meaningful feedback from teachers [9].

Developing SPS aligns with modern educational approaches that prioritize active, student-centered learning over traditional rote memorization [10]. Inquiry-Based Learning (IBL) is an effective pedagogical strategy that enhances SPS by actively engaging students in scientific inquiry, problem-solving, and hands-on experimentation. According to Gholam [11], IBL fosters critical thinking by requiring students to design and conduct experiments that address real-world problems, promoting a deeper understanding of scientific concepts. Kotsis [12] emphasize that IBL shifts students from passive learning to active engagement, allowing them to think independently and apply logical reasoning. Additionally, Schwartz et al. [13], highlight that IBL not only enhances theoretical knowledge but also develops students' practical investigation skills, preparing them for scientific careers. The collaborative aspect of IBL further strengthens SPS, as students work together to analyze data, discuss findings, and refine their understanding [14]. This approach cultivates essential scientific competencies such as communication, teamwork, and evidence-based decision-making, making IBL a valuable method for science education.

Despite the global recognition of Inquiry-Based Learning (IBL) as an effective teaching approach for developing SPS, research on its implementation and impact in Ethiopian secondary schools remains limited [15]. Moreover, despite the well-documented benefits of IBL, Ethiopian secondary schools face significant challenges in its effective implementation. A major barrier is the reliance of IBL on well-equipped laboratories and extensive materials, which many resource-limited schools in Ethiopia lack [16]. Additionally, the open-ended nature of IBL can lead to cognitive overload, making it difficult for students to independently design and conduct experiments without sufficient foundational knowledge [17]. Teacher resistance also hinders IBL adoption, as many educators prefer lecture-based methods and lack professional training in inquiry-based instruction [5]. Furthermore, student readiness and motivation significantly influence the success of IBL, as many learners struggle with self-directed inquiry without structured guidance[11]. These challenges similarly observed in other developing countries like Kenya and Nigeria [2,18]. Although some countries have made strides in adopting inquiry-based learning through reforms and teacher training, Ethiopia has seen limited progress in this area. Additionally, the gender disparities in academic achievement, as pointed out by Almasri [19], further underscore the need for comprehensive reforms in Ethiopian science education to address both the decline in students' academic performance and the underdevelopment of SPS. Addressing these challenges is essential to fully integrating IBL into Ethiopian science education and enhancing students' SPS development.

The insufficient development of SPS in Ethiopian secondary schools has significant implications for both individual students and society. Without well-developed SPS, students struggle to apply scientific reasoning and problem-solving techniques to real-world situations. This deficiency limits their ability to pursue careers in STEM fields, which are critical for national development [20]. Furthermore, the lack of strong SPS prevents students from effectively addressing pressing societal challenges. Public health issues such as disease outbreaks and sanitation problems require individuals with strong analytical and investigative skills to develop effective solutions [21]. Similarly, environmental conservation efforts depend on professionals who can conduct research, analyze ecological data, and implement sustainable practices [22]. Additionally, agricultural productivity, vital to Ethiopia's economy, benefits from scientific advancements that require a workforce with strong experimental and problem-solving skills [23]. Strengthening SPS in Ethiopian schools is therefore essential for fostering innovation and enhancing the country's ability to address both national and global challenges.

This study introduces the Guided Inquiry-Based Laboratory Experiments Enriched Instructional (GIBLEI) approach as a solution to challenges in resource-constrained educational settings. Unlike traditional Inquiry-Based Learning (IBL), GIBLEI integrates hands-on experimentation with guided inquiry, enhancing students' conceptual understanding, critical thinking, and problem-solving skills [24]. Rooted in constructivist learning theory, it fosters exploration and collaboration, allowing students to develop higher-order thinking skills such as hypothesis formulation, experimental design, and data interpretation through scaffolded instruction and cognitive apprenticeship [25]. GIBLEI addresses IBL challenges by incorporating locally available materials, virtual laboratory tools, and structured teacher facilitation to overcome resource constraints [6]. Additionally, it includes professional development programs to help teachers transition from traditional to inquiry-based instruction. By emphasizing collaborative learning and reflective discussions, GIBLEI enhances SPS while fostering autonomy, competence, and social interaction to improve motivation [26]. Furthermore, it strengthens communication and teamwork skills, linking laboratory experiences to real-world applications, and preparing students for STEM careers and practical problem-solving [26].

The GIBLEI approach is grounded in constructivism, cognitive apprenticeship, and self-determination theory, fostering active student engagement and inquiry-based learning.

Constructivism, as proposed by Piaget [27] and Yu, Hu, & Zhang [28], emphasizes knowledge construction through cognitive development and social interaction, respectively. The GIBLEI approach aligns with these principles by encouraging peer collaboration and hands-on experimentation [29]. Additionally, the cognitive apprenticeship model highlights the role of teachers as mentors who provide scaffolding and feedback to support students' development of SPS. Self-determination theory further reinforces the approach by promoting intrinsic motivation through autonomy, competence, and relatedness, allowing students to take ownership of their learning and engage deeply with scientific inquiry [26]. Through these theoretical foundations, GIBLEI enhances students' scientific understanding, problem-solving abilities, and motivation in resource-limited educational settings.

This study examines the practical application of the GIBLEI approach in resource-limited Ethiopian secondary schools to enhance science education. By addressing gaps in instructional practices, the research aims to improve students' SPS, thereby strengthening their preparedness for STEM careers and contributing to national development goals focused on innovation and technological advancement [30]. The findings inform educational reforms and teacher training programs, equipping educators with effective inquiry-based instructional strategies [5]. Furthermore, this study provides practical recommendations for improving teaching methodologies, curriculum development, and overall educational quality in under-resourced settings [31]. Thus, the main objective of this study was to examine the relative effectiveness of GIBLEI approach over traditional laboratory experiments enriched instructional approach in improving secondary schools biology students' SPS in North Shoa zone, Oromia Region, Ethiopia. To direct the investigation, the following null hypotheses were set forth and examined at the 0.05 level of significance:

*Ho*$_1$: The difference between the SPS post-test mean scores of students taught biology using GIBLEI and TLEIs is not significant.

*Ho*$_2$: The differences between the SPS pre-test and post-test mean scores in each group are not statistically significant.

*Ho*$_3$: The differences between the SPS post-test mean scores of male and female students' scores in each group are not statistically significant.

## 2. Materials and methods

### 2.1. Research design

In this study, a pre-test-post-test nonequivalent quasi-experimental research design was conducted to explore the effects of using GIBLEI approach on the SPS of secondary school students in biology. It is difficult to conduct true experiment in educational research for the reason that variables cannot be controlled fully and randomization is not easy [32]. This is because, students in schools are organized in to groups as classes at the beginning of each academic year by school administrators and will not be possible to assign the students randomly into groups as in true experiment [33]. However, the groups are considered to share similar characteristics. Wiseman [34] stated that when subjects are assigned to treatment groups without the application of randomization procedures, the design is described as a quasi-experimental design. The intervention was conducted from 25/03/2023–30/05/2023 Ethiopian academic year.

### 2.2. Samples and sampling technique

The study employed a multistage sampling technique to select comparable groups and minimize potential confounding variables. In the first stage, two public secondary schools, Fitche and Abdisa Aga, located in North Shoa Zone, Oromia Regional State,

Ethiopia, were purposively selected based on their similarities in educational resources and infrastructure, including laboratory facilities, library access, ICT infrastructure, and the qualifications and teaching experience of biology teachers. These similarities were deemed important to minimize school-level differences that could impact the study's findings. In the second stage, one class was randomly selected from each school. The class from Fitche Secondary School with 46 students (22 males and 24 females) was assigned as the experimental group (EG), while the class from Abdisa Aga Secondary School with 29 students (13 males and 16 females) was assigned as the control group (CG), resulting in a total of 75 Grade 10 students (35 boys and 40 girls). Additionally, two experienced biology teachers, one from each school, were purposively selected to maintain consistency in instructional quality.

Moreover, the topic "Food Making and Growth in Plants" was chosen for this study due to its critical role in plant biology and the frequent misconceptions students hold regarding key concepts like photosynthesis, nutrient transport, and plant responses. These biological processes are essential for understanding how plants grow, sustain life, and contribute to ecosystems, yet research indicates that many students misunderstand them. For instance, students commonly believe that plants obtain their food directly from the soil, a misconception referred to as the "soil-eater" belief [35]. They often fail to recognize that plants produce their own food through photosynthesis, which transforms light energy into chemical energy. Additionally, students frequently confuse photosynthesis and respiration, believing that plants only photosynthesize during the day and respire at night, rather than understanding that both processes occur continuously [35]. These misconceptions hinder students' grasp of core biological principles, such as energy flow in ecosystems and plant physiology [36].

Correcting misconceptions about photosynthesis is vital because it is a fundamental life-sustaining process that provides oxygen and organic matter necessary for nearly all living organisms. Moreover, photosynthesis is closely tied to pressing global challenges, such as climate change, food security, and sustainable agriculture [37]. Understanding how plants produce food is essential for scientific literacy, as it equips students with knowledge applicable to real-world issues. Studies have shown that inquiry-based learning strategies are particularly effective in helping students overcome misconceptions and achieve lasting conceptual change [38]. Therefore, selecting a topic prone to misconceptions allows the study to assess the impact of guided inquiry-based laboratory activities in improving students' understanding of difficult biological concepts. This choice is important for promoting students' academic success and preparing them to engage with complex scientific and environmental issues.

### 2.3. Data gathering instrument

In this study, science process skills (SPS) were assessed using a SPS essay test and assessment rubrics. The SPS essay test was adapted from Maradona [39] and modified by the researchers to align with the practical activities outlined in the Ethiopian grade 10 biology textbook. This test evaluated students' abilities in key SPS areas such as formulating hypotheses, designing experiments, interpreting data, and drawing conclusions. These areas are well-established indicators of proficiency in SPS and are commonly assessed at both high school and higher education levels [40]. The assessment rubrics used to evaluate the SPS essay tests were also adapted from Maradona [39]. These rubrics provided a structured framework for assessing student performance across the identified SPS variables The scoring range for the rubrics was from 0 to 4, with a maximum possible score of 16. The students individually conducted laboratory activities during the tests and completed both pre- and post-intervention essay tests. Their performance was evaluated using the rubrics.

**2.3.1. Validity of the SPS essay test.** In this study, the validity of an essay test designed to assess science process skills (SPS), along with its rubrics, was evaluated through content validity and inter-rater reliability. Content validity ensures whether the essay tests and the rubrics adequately cover the range of science process skills or not. Expert reviews are necessary to verify that all relevant skills are included and appropriately assessed. PhD students and two secondary school biology teachers evaluated whether the essay test was meant to assess SPS like formulating hypotheses, designing experiments, interpreting data, and drawing conclusions, and whether the rubrics ensured these skills were thoroughly evaluated. A study by Mirian [41] highlighted how expert panels can ensure comprehensive coverage of science process skills in assessments. Though inter-rater reliability is not a type of validity, ensuring consistent scoring among different raters using the rubric is essential for supporting the validity of the test results. For example, having multiple raters score essays and then calculating inter-rater reliability can ensure scoring consistency. Stolarova *et al.* [42] emphasized the importance of inter-rater reliability in their study by training raters and achieving high consistency in scoring. Thus, the validity of SPS (formulating hypotheses, designing experiments, interpreting data, and drawing conclusions) and rubrics was also ensured by the evaluation of a PhD student and the two secondary school biology teachers using the inter-rater reliability. In general, the study confirmed the validity of the essay test and rubrics for assessing SPS by combining content validity checks and ensuring scoring consistency through inter-rater reliability.

**2.3.2. Reliability of the instrument.** For the Science Process Skills (SPS) essay test, inter-rater reliability test was employed. On the first stage, the researcher and a PhD student/ colleague corrected the students' responses by using SPS rubrics. Then, the correlation of rated scores of students' performance by the researcher and the PhD student was tested by using intra-class correlation coefficient. The Intra-class Correlation Coefficient (ICC) is a statistic used to describe how strongly units in the same group resemble each other. It is widely used in the fields of psychology, education, and medicine to assess the reliability of measurements or ratings within groups or clusters. The ICC can be thought of as a measure of the consistency or reproducibility of quantitative measurements made by different observers measuring the same quantity. It essentially quantifies the degree to which individual scores are attributable to group membership, rather than to random error or individual differences [43].

Therefore, in this study Two-Way Mixed Effects Model (model 3) and average measures were employed. Moreover, absolute agreement ICC was used, since we want to measure how closer the values measured by each raters. Thus, it was revealed that the intra class correlation coefficient was 0.841, indicating acceptable reliability and suggesting that the tool can be implemented. The reliability was checked and ensured the test items were reliable and valid [44].

## 2.4. The treatment procedure

In the first stage of the intervention, the teacher and laboratory technician from the Experimental Group (EG) participated in a structured training program conducted by the researcher. This training focused on the instructional and training materials developed for the study, with a specific emphasis on the Guided Inquiry-Based Laboratory Experiments enriched Instruction (GIBLEI) approach. The training also included guidance on how to apply the 5E lesson plan format for daily lesson preparation. The training lasted for three days, with each session spanning 90 minutes. On the first day, the researcher provided a detailed explanation of the materials and instructional approach. The subsequent two days were dedicated to practical sessions, where the teacher and laboratory technician practiced implementing the GIBLEI approach in a classroom setting to ensure familiarity and confidence with the method.

Additionally, an orientation session was provided to the teacher and laboratory technician from the Control Group (CG) before the intervention began. This session outlined the aim and procedures of the research to ensure both groups were aware of their roles in the study. Following the training and orientation, a Science Process Skills (SPS) essay test was administered to students in both the EG and CG to establish a baseline for comparison. Instructional materials specifically designed for the EG were then distributed to the students. While the teachers' prior experience with inquiry-based methods was not formally assessed, the researchers addressed this potential variability through initial training and ongoing support throughout the study. The continuous guidance provided to both groups aimed to ensure that all teachers were adequately prepared to deliver the instructional approaches, thereby enhancing the consistency and reliability of the intervention outcomes.

Once all preliminary activities were completed, the intervention started. This intervention lasted about eight weeks (from 25/03/2023–30/05/2023) Ethiopian academic year. According to grade 10 biology syllabus, unit four required 24 periods to be completed. The lessons were delivered in alignment with the content distribution framework outlined in Table 1 below. The EG was taught using the GIBLEI approach, while the Traditional Laboratory Experiments enriched Instructional (TLEI) approach was used for the CG for the same topics (Table 1). The selected biology teachers, assisted by the researcher and lab technician, delivered each sub-topic according to the content distribution by preparing lesson plans for each sub-topic over the eight weeks. The eight-week duration of the intervention was aligned with practical considerations and prior research on implementing guided inquiry-based learning methods. Studies have shown that sustained exposure to inquiry-based instruction over an extended period is essential for students to internalize scientific practices and develop process skills [38]. The eight-week timeframe allowed sufficient time for students to engage deeply with the experimental processes and for teachers to effectively implement the instructional approach. This duration also aligned with the Ethiopian academic calendar, ensuring that the intervention covered a complete unit in the grade 10 biology syllabus.

The following procedures, adapted from Blanchard *et al.* [38], were applied to every GIBLEI. With the assistance of the lab technician, the selected teacher guided every step of the process.

**Stage1**. A week before each class, students received semi-structured problems from Unit 4 of the Grade 10 biology instructional materials prepared by the researchers. These problems were designed to guide students in thinking critically about the upcoming lab activities and experiments.

**Stage2.** Students independently researched experimental procedures relevant to the given problems. They planned their experiments, detailing each step they would take and

**Table 1. Weekly Content Distribution Framework.**

| Grade | Unit | Unit Topic | Week | Sub-topics |
|---|---|---|---|---|
| 10 | 4 | Food making and growth in plants | 1 | Organs of a flowering plant and the leaf |
| | | | 2 | Photosynthesis |
| | | | 3 | Photosynthesis |
| | | | 4 | Photosynthesis |
| | | | 5 | Transport in plants |
| | | | 6 | Transport in plants |
| | | | 7 | Response in plants |
| | | | 8 | Response in plants |

identifying the materials they needed. This phase allowed them to familiarize themselves with the scientific method and develop their problem-solving skills.

**Stage3.** Using the information gathered from their research, students outlined the layout of their experiments. This included formulating hypotheses, defining variables, and determining the steps to test their hypotheses.

**Stage4.** In groups, students shared their experimental designs with peers. They discussed each step of their planned experiments, the materials required, and the rationale behind their choices. This collaborative process encouraged critical thinking, peer feedback, and refinement of their experimental plans.

**Stage5.** The lab technician and teacher provided the necessary materials. Students then conducted their experiments according to their designs. During this phase, they meticulously recorded their observations and data, which would later be used to draw conclusions.

**Stage6.** In relation to the theoretical aspects of the unit, students presented their conclusions to the class. They explained how their observations and experimental data supported or refuted their hypotheses, linking practical findings to theoretical knowledge.

**Stage7.** Following the experiments, groups attempted to answer questions related to their experiments. These questions prompted deeper analysis and understanding of the experimental outcomes. The conclusions and responses were discussed in the classroom, fostering a comprehensive understanding of the topic.

**Stage8.** After each experiment, assessments were conducted to evaluate the students' understanding of both the theoretical concepts and the practical aspects of the experiments. These assessments helped determine how well the students grasped the scientific principles and methods covered in the lessons.

Furthermore, the selected teacher used to prepare daily lesson plans by using the phases of the 5E lesson plan format for the EG. Engage, explore, explain, elaborate, and evaluate are the phases. For each lesson, all the above steps were structured in to this format. The majority of the guided inquiry based laboratory activities were incorporated in to the exploration phase for the EG. The CG was taught the same topics by using the TLEI including lecture method and teacher demonstrations in which the students were spectators using students' textbooks. During the intervention, the researchers used to carry out meetings with the selected teachers at each week ends to discuss on how to prepare daily lesson plan and how to implement the activities of the intervention in the treatment. At the end of the interventions, SPS essay test was delivered to all the students in the two groups. Moreover, classroom observation was carried out by the researcher using checklist during the intervention to evaluate whether the instruction was delivered according to the plan or not. The researcher also used to take notes and spent some time to interact with the students, as a participant observer.

## 2.5. Data analysis

The data collected were first assessed to determine if they were parametric or non-parametric. SPSS version 22 was used for the analysis. For data to be considered parametric, it must meet the following criteria: normal distribution, homogeneity of variance, random selection of sample units, independent scores on the dependent variable, and data measured at least on an interval scale [45]. Normality was evaluated using skewness and kurtosis values, while homogeneity of variance was assessed with Levene's test. Skewness and kurtosis values should fall between -2 and +2 for data to be considered normally distributed [46]. In this study, the data were normally distributed. However, the variances of the post-test results for the experimental group (EG) and control group (CG) differed (Table 2). Thus, parametric tests were applied where the data met the normality and equal variance criteria, while non-parametric tests were used where these criteria were not met.

**Table 2. Normality and homogeneity tests for SPS scores.**

| Variables | Groups | Normality test | | Homogeneity test | | | |
|---|---|---|---|---|---|---|---|
| | | Skewness | Kurtosis | Levene's test | | | |
| | | | | F | df1 | df2 | P |
| SPS pretest | EG | .28 | −.33 | .72 | 1 | 73 | .399 |
| | CG | .48 | −.37 | | | | |
| SPS posttest | EG | .53 | −1.28 | 45.62 | 1 | 73 | .000 |
| | CG | .34 | −1.42 | | | | |
| | Male (EG) | .37 | −1.58 | .71 | 1 | 44 | .403 |
| | Female (EG) | .71 | −1.01 | | | | |
| | Male (CG) | .66 | −1.40 | .026 | 1 | 27 | .874 |
| | Female (CG) | .17 | −1.33 | | | | |

For the post-test mean score differences between the EG and CG, the study employed both Welch's t-test and Analysis of Covariance (ANCOVA) because these methods address distinct statistical requirements based on the characteristics of the data. Welch's t-test was applied to compare the post-test mean scores between the experimental group (EG) and control group (CG) due to the normal distribution of the data and unequal variances between the groups, as indicated by Levene's test [45]. This test is considered more robust than the standard independent t-test when homogeneity of variance is violated and is effective at controlling Type I errors in such situations[47]. It has also been shown to perform better than the Mann-Whitney U-test for normally distributed data with unequal variances [48]. Although no significant difference was found in pretest mean scores between the EG and CG, ANCOVA was used to control for residual differences in pretest scores and to improve the precision of post-test comparisons by adjusting for covariates [49]. ANCOVA is particularly suitable for quasi-experimental designs, where random assignment may not fully eliminate baseline differences, as it helps isolate the effect of the intervention on the post-test outcomes [50]. According to Field [45], ANCOVA enhances the statistical power of an analysis by reducing error variance, making it a valuable tool in educational research. To examine the differences between the post-test mean scores of male and female students in both groups, an independent samples t-test was used. The role of gender was a secondary analysis to determine whether the intervention effects varied between male and female students, providing additional insights into the study's findings. For comparing pretest and posttest mean scores within the EG, a Wilcoxon signed-rank test was applied, as the data were non-parametric (as determined by Kolmogorov-Smirnov and Shapiro-Wilk normality tests). Conversely, the paired samples t-test was used for the CG. Effect sizes were calculated using Cohen's d [51], ensuring a robust interpretation of the intervention's impact.

## 2.6. Ethical statement

The Research Review Ethics Committee of Hawassa University approved this study (Reference No. CNCS-REC029/22). Following approval, we received an official letter from the Department of Biology (Reference No. Bio/781/16), which was presented to the North Shoa Zone Education Office along with an explanation of the study's academic purpose. Upon receiving permission from the office, we conducted the study, obtaining written informed consent from the school directors, teachers, and students involved. These consent forms were archived alongside the collected data. Sample consent forms for school directors, teachers, and students are attached in Appendix C of this manuscript. Notably, the study included no minors, as all student participants were from the same age group within the society.

## 3. Results of the study

Table 3 shows the pretest mean scores for Science Process Skills (SPS) in both the Experimental Group (EG) and the Control Group (CG). The mean (M) score for the EG was 2.80 with a standard deviation (SD) of 1.75, indicating that, on average, students in this group had a higher SPS readiness score, with some variation around the mean. In contrast, the mean score for the CG was 2.27, with a standard deviation of 1.55, suggesting slightly lower average readiness but with comparable variability in scores.

To determine if the differences in mean scores between the two groups were statistically significant, an independent samples t-test was conducted. The results of the t-test (t = 1.32, p = .190) showed that the difference in mean scores between the groups was not statistically significant. The p-value of .190 is greater than the conventional significance level of 0.05, indicating that the observed difference is likely due to random variation rather than a true effect. This suggests that both groups had similar levels of SPS readiness before the introduction of Guided Inquiry-Based Laboratory Experiments Enriched Instruction (GIBLEI).

Table 4 presents the posttest mean scores for Science Process Skills (SPS) for both the EG and the CG. The EG students had a mean score of 5.30 with a standard deviation of 1.79, indicating a higher average SPS score with some variability among the students. In contrast, the CG students had a mean score of 1.69 with a higher standard deviation of 4.78, reflecting a lower average SPS score and greater variability. A Welch's t-test was conducted to compare the differences in the posttest SPS scores between the two groups. The results showed a significant difference (t = 4.54, p < .05), suggesting that the EG students outperformed the CG students in their posttest scores. Additionally, the large effect size ($\eta^2 = 0.82$) indicated that the intervention had a substantial impact on the EG students' SPS scores.

To further confirm these findings, an ANCOVA test was performed, controlling for pretest scores and other potential confounding factors. Table 5 indicates that the ANCOVA results revealed a significant effect of the intervention on SPS post-test scores, with an F-value of 12.36 and a p-value of .001 (p < 0.05). This confirms that the observed differences in SPS scores between the EG and CG were likely due to the Guided Inquiry-Based Laboratory Experiments Enriched Instruction (GIBLEI) intervention, rather than random chance. Together, the results from both the Welch's t-test and the ANCOVA provide strong evidence that the GIBLEI intervention significantly improved the SPS of the EG students.

**Table 3. Comparison of Pretest Science Process Skills (SPS) Mean Scores between the EG and CG.**

| Groups | N | Mean | SD | Df | T | p |
|---|---|---|---|---|---|---|
| Experimental group | 46 | 2.80 | 1.75 | 73 | 1.32 | .190 |
| Control group | 29 | 2.27 | 1.55 | | | |

**Table 4. Comparison of Posttest Science Process Skills (SPS) Mean Scores Between the EG and CG.**

| Groups | N | Mean | SD | T | P | η2 |
|---|---|---|---|---|---|---|
| EG | 46 | 5.30 | 4.78 | 4.54 | .000 | .82 |
| CG | 29 | 1.79 | 1.69 | | | |

**Table 5. ANCOVA Results for the SPS Post-Test Scores.**

| Test | Df | Mean Square | F | P |
|---|---|---|---|---|
| Post-test | 1 | 180.58 | 12.36 | .001 |

Consequently, the first null hypothesis (Ho1), which stated that there was no significant difference between the groups, is rejected.

Table 6 presents the results of a Wilcoxon signed-rank test comparing the pretest and posttest mean scores of Science Process Skills (SPS) within the EG. The pretest and posttest scores indicate a notable increase in SPS for the EG after receiving the Guided Inquiry-Based Laboratory Experiments enriched Instruction (GIBLEI) intervention. The test was conducted to account for the non-parametric nature of the data, which did not meet the assumptions of normality required for a paired t-test. The Wilcoxon signed-rank test revealed that 26 students in the EG had higher SPS posttest scores compared to their pretest scores (positive ranks), while 13 students showed a decrease in scores (negative ranks). The remaining 7 students had equal pretest and posttest scores. The analysis showed a statistically significant increase in SPS scores from pretest to posttest, with a z-value of -3.041 and a p-value of.002 (p < 0.05), indicating that the GIBLEI intervention had a substantial positive effect on the students' science process skills.

Table 7 shows the comparison of pretest and posttest mean scores for Science Process Skills (SPS) within the Control Group (CG) using a paired samples t-test. The mean SPS score for the pretest was M = 2.28 (SD = 1.56), while the posttest mean score was M = 1.79 (SD = 1.70). The slight decrease in the mean score indicates that the traditional laboratory experiment enriched instructional (TLEI) approach used in the CG did not lead to significant improvements in students' SPS performance. The paired samples t-test was conducted to assess whether the difference between the pretest and posttest mean scores was statistically significant. The results showed a t-value of 1.220 with a p-value of.233 (p > 0.05), indicating no significant improvement in SPS scores within the CG. This suggests that the control group's TLEI approach did not produce meaningful changes in students' SPS.

In contrast, the EG demonstrated a significant improvement in SPS after the intervention as confirmed by the previous analyses. These results support the rejection of the second null hypothesis (Ho2) for the EG, indicating that the Guided Inquiry-Based Laboratory Experiments Enriched Instruction (GIBLEI) intervention was effective in improving SPS in the experimental group, while no similar improvement was observed in the control group.

Table 8 presents the posttest mean scores for Science Process Skills (SPS) based on gender in both the EG and CG. In the EG, male students had a higher mean score (M = 6.32, SD = 4.88) compared to female students (M = 4.58, SD = 4.50), indicating some variation in performance. In contrast, in the CG, female students had a slightly higher mean score (M = 2.06, SD = 1.69) compared to male students (M = 1.46, SD = 1.71). To evaluate whether these observed

**Table 6. Comparison of Pretest and Posttest Science Process Skills (SPS) Mean Scores in the EG.**

| Test type | N | Row Mean | Row Total | Z | p |
|---|---|---|---|---|---|
| Neg. Ranks | 13 | 13.31 | 173 | -3.041 | .002 |
| Poz. Ranks | 26 | 23.35 | 607 | | |
| Equal | 7 | | | | |
| Total | 46 | | | | |

**Table 7. Comparison of Pretest and Posttest Science Process Skills (SPS) Mean Scores in the CG.**

| Test | N | Mean | SD | Df | T | p |
|---|---|---|---|---|---|---|
| Pretest | 29 | 2.2759 | 1.55601 | 28 | 1.220 | .233 |
| Post-test | 29 | 1.7931 | 1.69830 | | | |

**Table 8. Comparison of SPS Posttest Mean Scores Between Male and Female Students.**

| Groups | Gender | N | Mean | SD | Df | T | P |
|--------|--------|---|------|-----|-----|-----|-----|
| EG | M | 22 | 6.32 | 4.88 | 44 | 1.25 | .216 |
|  | F | 24 | 4.58 | 4.50 |  |  |  |
| CG | M | 13 | 1.46 | 1.71 | 27 | -.94 | .353 |
|  | F | 16 | 2.06 | 1.69 |  |  |  |

differences in SPS posttest scores between male and female students were statistically significant, independent samples t-tests were conducted. The results showed that in both groups, the differences were not statistically significant. In the EG, the t-value was 1.25 with a p-value of.216 (p > .05), and in the CG, the t-value was -.94 with a p-value of.353 (p > .05). These findings indicate that gender did not have a significant impact on SPS scores in either group.

The lack of significant gender differences supports the acceptance of the third null hypothesis (Ho3), which posited no significant difference in SPS posttest scores between male and female students. These results suggest that the Guided Inquiry-Based Laboratory Experiments Enriched Instruction (GIBLEI) intervention was equally effective for both male and female students in the EG, while TLEI in the CG did not produce gender-based differences in SPS performance.

## 4. Discussion of the findings

This study examined the impact of the Guided Inquiry-Based Laboratory Experiments Enriched Instructional (GIBLEI) approach on the development of Science Process Skills (SPS) among Grade 10 students in Ethiopian secondary schools. The findings clearly demonstrated that GIBLEI was significantly more effective in enhancing students' SPS compared to the Traditional Laboratory Experiments Enriched Instructional (TLEI) approach. The substantial improvement in SPS among students in the Experimental Group (EG) underscores the effectiveness of the GIBLEI intervention. The posttest mean score for the EG was significantly higher than that of the Control Group (CG). The Welch's t-test results (t = 4.54, p < .05) confirmed that this difference was statistically significant, with a large effect size ($\eta^2 = 0.82$). This indicates that approximately 82% of the variance in posttest SPS scores can be attributed to the GIBLEI intervention[52]. Additionally, the ANCOVA analysis, which controlled for pretest scores and potential confounding variables, further validated these findings [49]. The analysis confirmed that the improvements observed in the EG were directly linked to the GIBLEI approach (F = 12.36, p =.001), reinforcing the intervention's significant positive impact on students' SPS development.

The significant gains in SPS observed in the EG can be attributed to the active learning environment fostered by GIBLEI. Unlike traditional methods that focus on rote memorization and passive observation, GIBLEI emphasizes inquiry-based learning, where students actively engage in hypothesis formulation, experimentation, data interpretation, and conclusion drawing. This hands-on approach aligns with the constructivist theory of learning, which posits that knowledge is constructed through active engagement with the environment [27]. Additionally, the social constructivist theory emphasizes the importance of collaborative learning and guided discovery, both of which are integral components of GIBLEI. These theoretical underpinnings highlight how GIBLEI fosters critical thinking and problem-solving skills, which are essential for developing SPS [53]. The underlying mechanism behind the effectiveness of GIBLEI lies in its structured yet flexible framework that promotes student autonomy while ensuring appropriate scaffolding by teachers. The iterative cycles of hypothesis testing

and reflection embedded in GIBLEI provide students with opportunities to refine their scientific reasoning and experimental techniques. This continuous engagement with the scientific process cultivates higher-order thinking skills and fosters a deeper understanding of scientific concepts. Moreover, the integration of guided inquiry into laboratory activities ensures that students become active participants in their learning process, thereby enhancing their SPS.

The findings of this study are consistent with those of Nisa et al. [54], who reported that guided inquiry-based instruction enhances SPS and promotes higher-order thinking skills. Similarly, Wilson et al. [55] found that inquiry-based learning significantly improved both basic and integrated SPS in science education. These results are corroborated by Kotsis [12], who conducted a meta-analysis of inquiry-based science teaching and found substantial improvements in students' conceptual understanding and process skills. The integration of guided inquiry into the laboratory setting ensures that students are not only passive recipients of knowledge but active participants in their learning process, which is a critical factor in the development of SPS.

The Wilcoxon signed-rank test further demonstrated the effectiveness of GIBLEI in improving SPS over time. Within the EG, 26 students showed an increase in SPS scores from pretest to posttest, while only 13 students showed a decrease. The statistically significant z-value (-3.041, p = .002) indicates that the GIBLEI intervention led to a substantial improvement in students' SPS. These findings align with Lati et al. [56], who found that inquiry-based activities significantly enhance experimental and SPS skills. The continuous engagement and iterative learning process embedded in GIBLEI likely contributed to this sustained improvement.

An important aspect of this study is its exploration of gender differences in SPS outcomes. The results showed no significant gender-based differences in SPS performance within the EG, with male and female students achieving comparable posttest scores (t = 1.25, p = .216 for the EG; t = -0.94, p = .353 for the CG). This suggests that the GIBLEI approach is equally effective for both genders, promoting inclusive and equitable learning environments. These findings are consistent with Irma et al. [57] and El-Rabadi [58], who reported no significant gender differences in SPS outcomes following inquiry-based interventions. The study also aligns with the work of Uzezi and Zainab [59], who found that inquiry-based learning minimizes gender disparities in science education.

The lack of significant gender differences in SPS outcomes is particularly noteworthy in the Ethiopian context, where gender disparities in education remain a concern. The findings suggest that GIBLEI can serve as a tool to promote gender equity in science classrooms by providing equal opportunities for both male and female students to develop critical SPS. This aligns with global educational priorities, such as the United Nations Sustainable Development Goal 4, which emphasizes inclusive and equitable quality education for all. Additionally, this finding underscores the potential of GIBLEI to contribute to closing gender gaps in STEM fields, a persistent challenge in both developing and developed countries [60].

The relevance of GIBLEI extends beyond the Ethiopian context to other resource-limited settings worldwide. Studies from rural schools in India, sub-Saharan Africa, and other low-income regions have demonstrated the feasibility and effectiveness of low-resource adaptations of inquiry-based learning [61]. In these contexts, teachers have successfully implemented guided inquiry approaches using low-cost materials and virtual laboratory simulations, achieving significant improvements in students' SPS and conceptual understanding. Asare et al. [62] suggest that virtual laboratories can serve as a practical alternative in resource-constrained settings, providing students with opportunities to engage in simulated experiments. The use of virtual labs can complement physical laboratory activities, enabling students to practice inquiry-based skills even when access to physical resources is limited. Teacher training programs should

also focus on equipping educators with strategies for implementing inquiry-based learning in low-resource environments, ensuring that all students, regardless of their geographic location or socioeconomic status, have access to high-quality science education.

## 5. Conclusion and implications

The study demonstrates that the Guided Inquiry-Based Laboratory Experiments enriched Instructional (GIBLEI) approach significantly enhances science process skills (SPS) in secondary school students compared to the Traditional Laboratory Experiments enriched Instructional (TLEI) approach, particularly in biology topics such as food making and plant growth. Posttest results show that students in the control group (CG), who were taught using TLEI, did not perform as well as students in the experimental group (EG), who were taught using the GIBLEI approach. Furthermore, students in the EG showed improved performance in their SPS posttests compared to their pretest results, suggesting a positive effect of the intervention. The data also reveal that GIBLEI benefits all students equally, as evidenced by similar improvements in SPS for both male and female students in the EG. This indicates that the GIBLEI approach is not gender-biased and positively impacts all students. In conclusion, the GIBLEI approach is more effective than the TLEI approach in enhancing the SPS of secondary school students in biology.

The findings of this study on the GIBLEI approach carry significant implications for advancing science education both in Ethiopia and beyond. First, the results advocate for a transformative shift in curriculum design towards active learning methods that prioritize hands-on, inquiry-driven activities rather than traditional rote learning practices. Such a shift fosters the development of essential scientific process skills (SPS), critical thinking, and problem-solving abilities while enhancing students' overall scientific literacy.

Second, the study emphasizes the need for innovative instructional practices that go beyond conventional teaching methods. Successfully implementing GIBLEI requires comprehensive teacher training programs focused on inquiry-based pedagogy. These programs should empower educators to design and facilitate guided inquiry experiments, make effective use of low-cost materials, and incorporate virtual lab simulations where practical. Continuous professional development is crucial to ensure the sustainability and effectiveness of these instructional strategies.

Lastly, the findings provide practical recommendations for adapting the GIBLEI approach in resource-limited settings by addressing key challenges through innovative and adaptable strategies. Utilizing locally available materials allows schools to conduct cost-effective laboratory experiments without compromising the quality of hands-on learning, making science education more accessible in low-resource environments. Incorporating virtual laboratory tools further complements physical experiments by offering interactive simulations of complex scientific processes that may not be feasible with limited equipment. Promoting unity-based science projects encourages students to apply their scientific knowledge to solve local issues, thereby reinforcing learning while fostering a sense of responsibility and connection to their surroundings. Addressing gender disparities in science education is crucial for creating inclusive learning environments. This can be achieved through gender-sensitive teaching practices and initiatives that encourage both boys and girls to actively participate in inquiry-based learning. By implementing these strategies, GIBLEI not only improves educational outcomes in local contexts but also offers scalable, adaptable solutions that can enhance science education globally, particularly in regions facing similar challenges. The approach ensures that quality science education is attainable even in resource-constrained settings by emphasizing innovative teaching methods and equitable access for all learners.

## Limitations

This study encountered some limitations that could have influenced the research outcomes. Variations in resource availability, teacher-related differences, and the relatively small sample size were key challenges. The research was conducted in natural school environments where class assignments were organized by school administrators. While this reflects real-world educational conditions, it may limit the applicability of the findings to other contexts with different administrative structures. Furthermore, the sample size, comprising 75 Grade 10 students from two schools, may have restricted the diversity of perspectives captured. As a result, the findings may not fully represent broader student populations, making it necessary to interpret the results with caution when generalizing them to different regions or more diverse educational settings.

Despite these limitations, the study provides important insights into the effectiveness of guided inquiry-based instruction in biology education. The relatively small sample size reduces the statistical power of the study and may limit its ability to capture the full range of academic performance and learning styles found in larger populations. However, the study highlights the potential benefits of this instructional approach, particularly in schools with similar educational environments. To strengthen future research, strategies such as standardizing educational resources, improving teacher training, implementing randomized controlled trials, conducting longitudinal studies, and fostering collaborations with educational stakeholders can be adopted. These measures would help address the identified limitations and contribute to a broader understanding of effective teaching practices that enhance students' science process skills and learning outcomes in biology education.

## Supporting information

**SI File  Appendix-A. Science Process Skills (SPS) Essay Test.** This appendix includes the SPS essay test provided to students, detailing sample experimental topics, instructions, and essay questions designed to assess students' science process skills in biology. SI Appendix-B). SPS Essay Test RUBRIC. This rubric outlines the scoring criteria for evaluating students' responses to the SPS essay test, including indicators for formulating hypotheses, designing experiments, interpreting data, and drawing conclusions.
(DOCX)

**S2 File.  SPS Research Data.**
(DOCX)

## Acknowledgment

We are grateful to the department of Biology, Hawassa University, for unlimited material and official support.

## Author contributions

**Conceptualization:** Beyene Dobo Bono, Samuel Assefa Zinabu.

**Data curation:** Ashebir Mekonnen Chengere, Beyene Dobo Bono.

**Formal analysis:** Ashebir Mekonnen Chengere.

**Investigation:** Ashebir Mekonnen Chengere, Samuel Assefa Zinabu, Kedir Woliy Jilo.

**Methodology:** Ashebir Mekonnen Chengere, Beyene Dobo Bono, Samuel Assefa Zinabu, Kedir Woliy Jilo.

**Project administration:** Ashebir Mekonnen Chengere.

**Resources:** Kedir Woliy Jilo.

**Supervision:** Beyene Dobo Bono, Samuel Assefa Zinabu, Kedir Woliy Jilo.

**Validation:** Ashebir Mekonnen Chengere, Beyene Dobo Bono, Samuel Assefa Zinabu, Kedir Woliy Jilo.

**Visualization:** Ashebir Mekonnen Chengere.

**Writing – original draft:** Ashebir Mekonnen Chengere.

**Writing – review & editing:** Ashebir Mekonnen Chengere.

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
