## [Decision Letter · Decision Letter 0]

27 Dec 2024

PONE-D-24-48585Enhancing Secondary School Students’ Science Process Skills through Guided Inquiry-based Laboratory activities in BiologyPLOS ONE

Dear Dr. Chengere,

Thank you for submitting your manuscript to PLOS ONE. After careful consideration, we feel that it has merit but does not fully meet PLOS ONE’s publication criteria as it currently stands. Therefore, we invite you to submit a revised version of the manuscript that addresses the points raised during the review process.

Below, I have outlined the revisions needed:

**A. Abstract**

Structure: Rearrange to present background information first, followed by the research objective for better logical flow.Clarity: Explicitly link the research aim to the educational challenges, such as traditional teaching methods and resource constraints.Keywords: Include "secondary school students" to improve discoverability.

**B. Introduction**

State-of-the-Art: Expand the discussion of previous studies to articulate the research gap and novelty of the current work more convincingly.Problem Statement: Clearly frame the problem and connect the study's significance to broader implications for Ethiopian education and global education research.Justification: Provide a rationale for conducting the study, particularly given existing findings that inquiry-based learning improves SPS. Explain how the current work builds on or differs from past studies.Sample Size: Acknowledge the small sample size and its implications for statistical power and generalizability.

**C. Materials and Methods**

Research Design:Justify the selection of the biology topic.Clarify and provide evidence that the groups had similar characteristics, specifying which characteristics were compared.Explain the use of two statistical analyses (Welch’s t-test and ANCOVA) and why one consistent method was not chosen.
Gender Analysis:Clarify the role of gender in the study (e.g., independent variable or secondary analysis).If gender is a primary factor, justify why a two-way ANCOVA was not used.
Intervention Details:Explain the rationale for the eight-week intervention duration and discuss its alignment with prior research or practical considerations.Indicate whether teachers’ prior experience with inquiry-based methods was assessed, as this could affect the intervention’s outcomes.

**D. Results**

Presentation:Provide clear captions for tables that include descriptions, statistical tests used, and significance levels (e.g., p < 0.05).
Sample Size: Acknowledge how the small sample size may limit the diversity of perspectives and findings.

**E. Discussion**

Depth and References:Expand the discussion to explore why the intervention was effective, particularly the role of active learning and student engagement in fostering SPS.Provide more references to support interpretations and findings.
Mechanisms and Implications: Analyze the underlying mechanisms driving the significant effects on students’ competencies and explore the broader implications for curriculum design.Global Relevance: Broaden the discussion to include examples of similar approaches in resource-limited settings or other international contexts.Practical Recommendations: Suggest strategies for adapting GIBLEI for resource-constrained environments.

**F. General Comments**

Formatting: Ensure consistent formatting, including spacing, alignment, and adherence to journal guidelines.Grammar and Style: Address minor grammatical issues to improve clarity and flow throughout the manuscript.

We look forward to receiving your revised manuscript.

Kind regards,

Mc Rollyn Daquiado Vallespin

Academic Editor

PLOS ONE

Reviewers' comments:

Reviewer's Responses to Questions

**Comments to the Author**

1. Is the manuscript technically sound, and do the data support the conclusions?

Reviewer #1: No

Reviewer #2: Yes

Reviewer #3: Yes

2. Has the statistical analysis been performed appropriately and rigorously?

Reviewer #1: Yes

Reviewer #2: I Don't Know

Reviewer #3: Yes

3. Have the authors made all data underlying the findings in their manuscript fully available?

Reviewer #1: Yes

Reviewer #2: Yes

Reviewer #3: Yes

4. Is the manuscript presented in an intelligible fashion and written in standard English?

Reviewer #1: No

Reviewer #2: Yes

Reviewer #3: Yes

5. Review Comments to the Author

Reviewer #1: Abstract:

The first sentence of the abstract presents the research objective, while the second and third sentences provide the research background. It would be more effective to present the background information first, followed by the objective.

Introduction:

The state-of-the-art section is weak due to the limited discussion of previous studies. As a result, the articulation of the research gap and the novelty is not strong enough.

Methods:

The research design is overly simplistic, involving only one independent variable, one dependent variable, and two groups.

In the research design section, provide justification or evidence that both groups have similar characteristics and clarify what specific characteristics are deemed similar.

In the data analysis section, explain why you used two different analyses (Welch's t-test and ANCOVA) to compare the performance of students in the two groups instead of opting for one consistent method.

Regarding gender: How is it positioned in the study? Why did you compare the performance of male and female students? If gender is treated as an independent variable, why did you not employ a two-way ANCOVA?

Discussion:

The discussion is too brief, comprising only two paragraphs. It is also inadequately supported by references, with only one scientific citation in the first paragraph.

The discussion primarily focuses on interpreting the analytical results and comparing findings with previous studies. The manuscript lacks a deeper exploration of why the treatment yielded significant effects on students’ competencies. A more thorough analysis and discussion of the mechanisms and implications of the findings are necessary.

Reviewer #2: Thank you for selecting me to review the manuscript titled, "Enhancing Secondary School Students’ Science Process Skills through Guided Inquiry-Based Laboratory Activities in Biology." The article is well-written and presents an important contribution to the field of science education. However, I recommend the following revisions to strengthen the manuscript:

Abstract

1. Refine the aim for greater clarity by explicitly linking it to the educational challenges mentioned (e.g., resource constraints, traditional methods). For example:

"This study aims to evaluate the effectiveness of a Guided Inquiry-Based Laboratory Experiments Enriched Instructional (GIBLEI) approach in overcoming traditional teaching limitations and enhancing science process skills (SPS) among Ethiopian secondary school biology students."

2. Add "secondary school students" to the list of keywords to improve discoverability.

Introduction

1. Clearly frame the problem statement and expand the significance of the study to connect the findings to broader implications for Ethiopian education and global education research.

2. Acknowledge the small sample size (N = 75, with only 29 in the control group) and discuss its implications for statistical power and generalizability.

Materials and Methods

1. Indicate whether the teachers' prior experience with inquiry-based methods was assessed, as this could influence the intervention's effectiveness.

2. Justify why eight weeks was chosen as the intervention duration and discuss whether this aligns with previous research or practical constraints.

Results

1. Provide clear and descriptive captions for all tables, ensuring each includes:

o A brief explanation of the table's content.

o The statistical test performed (e.g., t-test, ANCOVA).

o The significance level (e.g., p<0.05p < 0.05).

Discussion

1. Delve deeper into why GIBLEI was particularly effective, exploring the role of active learning or student engagement in fostering SPS.

2. Discuss how GIBLEI could be adapted for resource-limited settings, providing practical recommendations for implementation.

General Comments

1. Ensure consistency in formatting, including spacing between brackets and words.

2. Address minor grammatical issues throughout the manuscript to improve clarity and flow.

Reviewer #3: 1. “When laboratory experiments are used as an instructional strategy and students are actively engaged, most studies have found a significant improvement in students' achievement and SPS (9). Therefore, active learning techniques, such as inquiry-based learning, should replace traditional lecturing and cookbook-style laboratories”

If this has already been established, then what is the rational or basis for the current work?

2. Please provide justification for the selected biology topic

6. PLOS authors have the option to publish the peer review history of their article (what does this mean? ). If published, this will include your full peer review and any attached files.

**Do you want your identity to be public for this peer review?** For information about this choice, including consent withdrawal, please see our Privacy Policy .

Reviewer #1: **Yes: ** Ahmad Fauzi

Reviewer #2: **Yes: ** Zhikal Omar Khudhur

Reviewer #3: **Yes: ** Yeboah Kwaku Opoku

---

## [Author Response · Author response to Decision Letter 1]

7 Feb 2025

Date: January 20, 2025

Dear Mc Rollyn Daquiado Vallespin (Academic Editor)

PLOS ONE

Subject: Report on the incorporation of comments and suggestions

Following our submission of the manuscript for publication we have received invaluable comments and suggestions that have improved the quality of the manuscript. We acknowledge the contributions of the editorial team and the reviewers for valuing our work and allowing us to improve the quality standard of the manuscript. We have incorporated all the comments/suggestions provided by the reviewers and the response on how we incorporated the comments is annexed below. Thank you again for your consideration!

With Best Regards

Content Suggestions/comments Response (ways how they are addressed)

Abstract Reviewer 1:

i. The first sentence of the abstract presents the research objective, while the second and third sentences provide the research background. It would be more effective to present the background information first, followed by the objective.

Reviewer 2:

i. Refine the aim for greater clarity by explicitly linking it to the educational challenges mentioned (e.g., resource constraints, traditional methods). For example: "This study aims to evaluate the effectiveness of a Guided Inquiry-Based Laboratory Experiments Enriched Instructional (GIBLEI) approach in overcoming traditional teaching limitations and enhancing science process skills (SPS) among Ethiopian secondary school biology students."

ii. Add "secondary school students" to the list of keywords to improve discoverability. i. We completely accepted the comments, and we revised the abstract based on the given comments. We have highlighted the revised text in bright green color.

i. We accepted the comment and tried to clarify the research aim by linking it to the educational challenges. Moreover, we highlighted the revision with dark yellow color.

ii. We accepted the suggestion and the phrase "secondary school students" in to key words. The change was highlighted with green color.

Introduction Reviewer 1:

i. The state-of-the-art section is weak due to the limited discussion of previous studies. As a result, the articulation of the research gap and the novelty is not strong enough.

Reviewer 2:

i. Clearly frame the problem statement and expand the significance of the study to connect the findings to broader implications for Ethiopian education and global education research.

ii. Acknowledge the small sample size (N = 75, with only 29 in the control group) and discuss its implications for statistical power and generalizability.

Provide a rationale for conducting the study, particularly given existing findings that inquiry-based learning improves SPS. Explain how the current work builds on or differs from past studies.

Reviewer 3:

i. Active learning techniques, such as inquiry-based learning, should replace traditional lecturing and cookbook-style laboratories. If this has already been established, then what is the rational or basis for the current work? By accepting the suggestions:

i. We have tried to incorporate enough previous relevant studies in to the introduction part to enhance the articulation of the research gap and strengthen the novelty of the research. For instance, the introduction provides a solid overview of the literature related to Science Process Skills (SPS), inquiry-based learning (IBL), and the challenges faced in Ethiopian secondary schools. Moreover, the discussion expanded to clarify the specific research gap and the novelty of the current work. The purple color indicates some of the modification we have made.

i. We have tried to clarify the problem statement and the significance of the study in the introduction section. The blue color indicates some of the modification we have made.

ii. We accepted the comment and acknowledged in the small sample size in the limitation sections. The change was highlighted with light green color.

We completely accepted the comment and included the rationale for conducting the study and how the current work builds on or differs from past studies. Some of the changes were highlighted with yellow color.

i. We accepted the comment and revised the content by completely removing the vague sentence.

Methods

Reviewer 1:

i. In the research design section, provide justification or evidence that both groups have similar characteristics and clarify what specific characteristics are deemed similar.

ii. In the data analysis section, explain why the two different analyses (Welch's t-test and ANCOVA) to compare the performance of students in the two groups instead of opting for one consistent method.

iii. Regarding gender: How is it positioned in the study? Why did you compare the performance of male and female students? If gender is treated as an independent variable, why did you not employ a two-way ANCOVA?

Reviewer 2:

i. Indicate whether the teachers' prior experience with inquiry-based methods was assessed, as this could influence the intervention's effectiveness.

ii. Justify why eight weeks was chosen as the intervention duration and discuss whether this aligns with previous research or practical constraints. By accepting the suggestions/Comments:

i. We included the features of the schools and the profiles of teachers in the sampling section and highlighted with yellow color (These schools were chosen based on their similarity in facilities, including laboratory chemicals and equipment, library resources, ICT infrastructure, and the qualifications and experience of teachers).

ii. We accepted the comment and we gave reason for simultaneous use of Welch's t-test and ANCOVA in the data analysis section. We also highlighted the changes with turquoises color.

iii. In this study, the role of gender was a secondary analysis to determine whether the intervention effects varied between male and female students, providing additional insights into the study’s findings. It is highlighted with pink color in the data analysis section.

i. The teachers’ prior experience with inquiry-based methods was not formally assessed. However, the researchers addressed this potential variability through initial training and ongoing support throughout the study. The continuous guidance provided to both groups aimed to ensure that all teachers were adequately prepared to deliver the instructional approaches, thereby enhancing the consistency and reliability of the intervention outcomes. This was included in the treatment procedure section and highlighted with pink color.

ii. We accepted the comment and we have included the justification in the treatment procedure section. The changes were highlighted with dark yellow color.

Results Reviewer 2:

i. Provide clear and descriptive captions for all tables, ensuring each includes:

• A brief explanation of the table's content.

• The statistical test performed (e.g., t-test, ANCOVA).

• The significance level (e.g., p<0.05p < 0.05).

i. We accepted the comments and we provided clear and descriptive captions for all the tables according to the comments.

Discussion Reviewer 1:

a. The discussion is too brief, comprising only two paragraphs. It is also inadequately supported by references, with only one scientific citation in the first paragraph.

b. The discussion primarily focuses on interpreting the analytical results and comparing findings with previous studies.

c. The manuscript lacks a deeper exploration of why the treatment yielded significant effects on students’ competencies.

d. A more thorough analysis and discussion of the mechanisms of the findings are necessary.

e. The implications of the findings are also important.

Reviewer 2:

f. Delve deeper into why GIBLEI was particularly effective, exploring the role of active learning or student engagement in fostering SPS.

g. Discuss how GIBLEI could be adapted for resource-limited settings

h. Provide practical recommendations for implementation.

i. We accepted the comments. Moreover, we revised and supported the discussion with more relevant references.

ii. We completely accepted the comment. Thus;

a. The treatment yielded significant effects on students’ competencies due to the nature the GIBLEI. This was highlighted with bright green color in the discussion section.

b. We also added the underlying mechanism behind the effectiveness of GIBLEI in discussion sections. It was also highlighted with blue color in the discussion sections.

c. Moreover, the implications of the findings were also included in the conclusion and recommendation sections. The implications were highlighted with teal color.

i. We completely accepted the comment and incorporated the reason why GIBLEI approach was effective in developing students SPS. This was highlighted with bright green color.

ii. The mechanisms by which the GIBLEI could be adapted for resource-limited settings was discussed in the last paragraph of the discussion section and highlighted with pink color.

iii. The practical recommendations for implementation of the GIBLEI was provided in the last paragraph of the conclusion and recommendation section and highlighted with yellow color.

---

## [Editor Report · Decision Letter 1]

23 Feb 2025

Enhancing Secondary School Students’ Science Process Skills through Guided Inquiry-based Laboratory activities in Biology

PONE-D-24-48585R1

Dear Mr/Ms. Chengere,

We’re pleased to inform you that your manuscript has been judged scientifically suitable for publication and will be formally accepted for publication once it meets all outstanding technical requirements.

Kind regards,

Mc Rollyn Daquiado Vallespin

Academic Editor

PLOS ONE
---

## [Editor Report · Acceptance letter]

PONE-D-24-48585R1

PLOS ONE

Dear Dr. Chengere,

I'm pleased to inform you that your manuscript has been deemed suitable for publication in PLOS ONE. Congratulations! Your manuscript is now being handed over to our production team.

Kind regards,

on behalf of

Dr. Mc Rollyn Daquiado Vallespin

Academic Editor

PLOS ONE